TECHNICAL RELEASE

# HTGQC and shinyHTGQC: an R package and shinyR application for quality controls of HTG EDGE-seq protocols

Lodovico Terzi di Bergamo[1,2,3,*], Francesca Guidetti[4,5], Davide Rossi[1], Francesco Bertoni[4] and Luciano Cascione[2,4]

1 Laboratory of Experimental Hematology, Institute of Oncology Research, Bellinzona, Switzerland
2 Bioinformatics Core Unit, Swiss Institute of Bioinformatics, Bellinzona, Switzerland
3 Department of Health Science and Technology, Swiss Federal Institute of Technology (ETH Zürich), Zurich, Switzerland
4 Lymphoma Genomics unit, Institute of Oncology Research, Bellinzona, Switzerland
5 Faculty of Biomedical Sciences, Università della Svizzera Italiana, Lugano, Switzerland

## ABSTRACT

Extraction-free HTG EdgeSeq protocols are used to profile sets of genes and measure their expression. Thus, these protocols are frequently used to characterise tumours and their microenvironments. However, although positive and control genes are provided, little indication is given concerning the assessment of the technical success of each sample within the sequencing run. We developed HTGQC, an R package for the quality control of HTG EdgeSeq protocols. Additionally, shinyHTGQC is a shiny application for users without computing knowledge, providing an easy-to-use interface for data quality control and visualisation. Quality checks can be performed on the raw sequencing outputs, and samples are flagged as FAIL or ALERT based on the expression levels of the positive and negative control genes.

**Availability & Implementation:** The code is freely available at https://github.com/LodovicoTerzi/HTGQC (R package) and https://lodovico.shinyapps.io/shinyHTGQC/ (shiny application), including test datasets.

**Subjects** Software and Workflows, Bioinformatics, Transcriptomics

Submitted: 07 October 2022

* Corresponding author. E-mail: lodovico.terzi@ior.usi.ch

Preprint submitted at https://doi.org/10.5281/zenodo.7351234

## BACKGROUND

The advances in high-throughput sequencing have demonstrated the importance of gene expression profiling for characterizing and identifying subgroups in various cancer types and their response to therapy [1–3]. While total RNA sequencing helps highlight genome-wide differences among patients for survival prediction [4] and discover novel uncharacterised transcripts [5], the targeted sequencing of mRNA expression signatures helps analyse custom-made panels of genes of interest [6]. This includes cancer-specific genes for subtype classification or immune-related genes for deconvoluting the microenvironment composition efficiently and cost-effectively [7].

HTG Molecular Diagnostics is a life science company that aims at accelerating precision medicine by a multitude of RNA-based profiles. Of particular interest are the targeted panels of transcriptome profiles, including the Diffuse Large B-Cell Lymphoma Cell of Origin (COO) assay for COO classification [1], and a number of panels for studying the

tumour and its microenvironment composition (Oncology Biomarker Panel and Precision Immuno-Oncology Panel) [8, 9]. Formalin-Fixed Paraffin-Embedded (FFPE) is a widely used method for preserving tissue specimens that allows the preservation and easy storage of biological samples for years. Therefore, the extraction-free HTG EdgeSeq panels are optimised for FFPE samples. Allowing the reliable quantification of RNA expressions using FFPE samples, these panels are of great interest to the biological world [10].

Although the wet lab protocols are highly standardised, little indication is given concerning the analysis of the resulting sequencing data. A fundamental step is the assessment of the success of the sequencing runs and the quality of individual samples [11]. In this regard, HTG EdgeSeq panels provide four positive controls (spike-ins) and four negative controls (non-human genes). Before proceeding with the data analysis, normalisation and quality controls are critical for such custom-made panels.

The amount of data created by high-throughput machines is constantly increasing; however, the tools allowing biologists to independently perform basic data cleaning, visualisation and analysis are still mostly inadequate.

Here, we introduce HTGQC and shinyHTGQC, the first tool, to our knowledge, for the quality control of HTG EdgeSeq protocols. Since no well-defined pipeline is currently advised or in use for the analysis of this type of data, our standardised quality control tool can be considered as a building block and starting point for downstream applications, such as differential expression and gene set enrichment analyses.

## METHODS

We provide an easy-to-use tool, along with clean visualisations of the quality control, in the form of an R package (RRID:SCR_001905). Moreover, a Shiny application (RRID:SCR_001626) has been created to facilitate quality controls and data analysis for those unfamiliar with the R language [12] (Figure 1).  The R package HTGQC is composed of two functions. The first one, *readHTG* (), processes the output from the EdgeSeq machine (a .xlsx file) with no need for preprocessing or manual curation. The main function, *qualityCheck* (), performs the quality control of the samples using both positive and negative controls.

### Positive controls

The percentage of reads allocated to the positive controls over the total library size is plotted. A table is produced listing the samples not passing the quality control, with a threshold of 40% set for the failures. Based on previous analyses, this percentage tends to be in the range of 0–5% for good-quality samples. Therefore, we added an ALERT flag when the percentage of reads allocated to positive controls exceeds 10%.

### Negative controls

Gene counts are normalised in counts per million (cpm), and the mean cpm of the negative controls is calculated for each sample. To obtain the deviance from the expected value (Δ) for each sample $i = \{1, \dots, N\}$, each of these values is subtracted from the average cpm across all samples. For genes $j = \{1, \dots, M\}$, and if $X_{i,j}$ is the observed expression for patient $i$ and gene $j$,

$$\Delta_i = \mathrm{CPM}_i - \frac{\sum_1^N \mathrm{CPM}_j}{N}$$

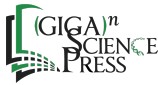

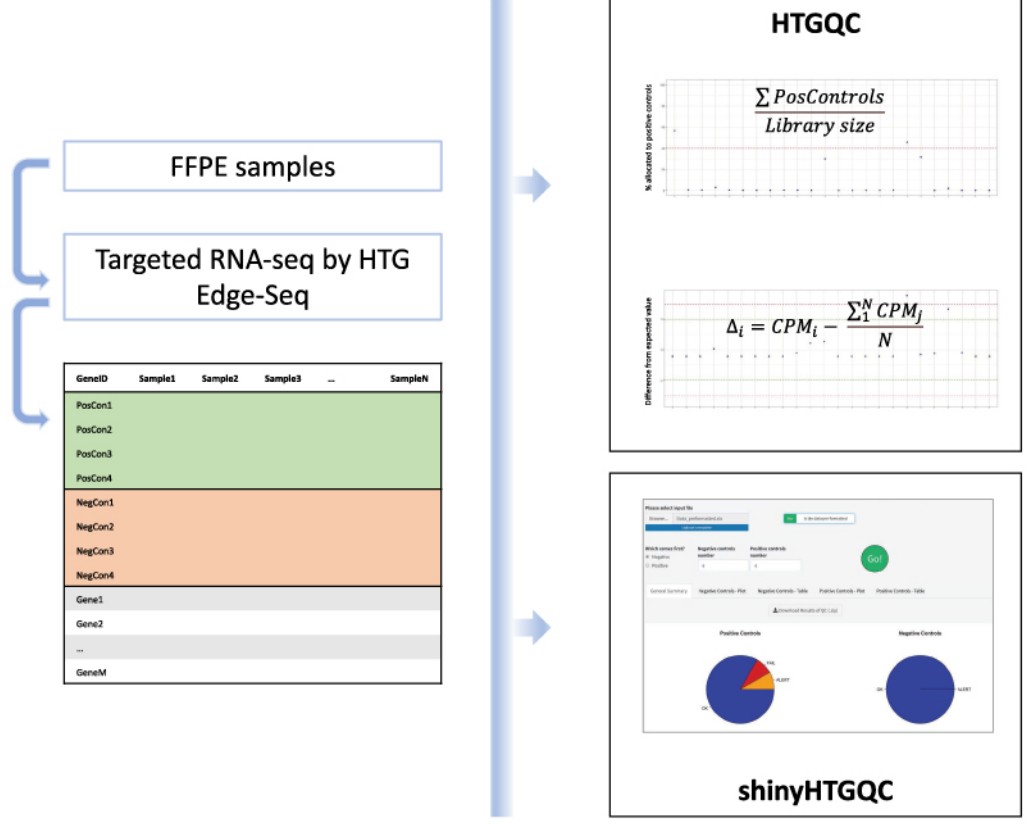

**Figure 1. Workflow of HTGQC and shinyHTGQC.**
Gene expression data from HTG EdgeSeq Protocols are analysed for quality control of the sequencing run. HTGQC uses the positive and negative controls provided in these datasets for assessing failures. shinyHTGQC is a web-based tool for quality assessment and visualisation.

where

$$\mathrm{CPM}_i = \frac{\sum_1^M \mathrm{cpm}(X_{i,j})}{M}$$

In the case of a complete failure of all the samples in the run, there is a possibility that all deviances will be within the $\pm 2^*$ standard deviation range since all the samples have a high number of reads allocated to negative controls. We have therefore added a second filter in which, similarly to the positive controls, a sample fails the quality control if the percentage of reads allocated to the negative controls exceeds 10% of its library size.

Finally, each patient's quality control is flagged ALERT when any of the two filters is ALERT, while it is flagged FAIL when any filter is FAIL.

## Output

The results are two tables and two plots. The tables report the success and failure samples, while the plots represent the percentage of reads allocated to the negative controls and the deviance from the expected value for the negative controls. If the input data was not pre-processed, an additional option is available for downloading the cleaned data, ready to be used for additional analyses.

The package has been trained and validated for the Oncology Biomarker and Precision Immuno-Oncology HTG panels. However, some options can be customised (i.e., the number of positive and control genes) to allow further applications.

## Shiny application

The web-based shiny application shinyHTGQC has been implemented using the same concept for the quality controls above. This tool allows researchers with no knowledge of the R language to perform the same analyses.

shinyHTGQC only requires the user to upload the input data files and specify the control genes to be used. The application performs the quality check and provides the user with the possibility of visualizing and downloading the results in a user-friendly way.

## Data normalisation and analysis

Since shinyHTGQC will be used by researchers with little or no knowledge of bioinformatics, we also added features for visualising the effect of the quality control analysis.

The user is asked to select a normalisation method between cpm and variance stabilizing transformation (vst). The default selection is cpm, as suggested by the authors.

A principal component analysis (PCA) is plotted in the 'Analysis' tab of the shiny application, along with a heatmap visualisation of the samples (column) and genes (rows).

In order to visualise the effect of removing patients flagged as FAIL/ALERT by the quality control, a further option allows the users to exclude these patients from the graphical representations.

The user can specify an annotation file to specify the different groupings of the samples. In this case, the PCA will be coloured and the heatmap annotated accordingly.

## AVAILABILITY OF SUPPORTING SOURCE CODE AND REQUIREMENTS

- Project Name: HTGQC; shinyHTGQC
- Project home page: https://github.com/LodovicoTerzi/HTGQC; https://lodovico.shinyapps.io/shinyHTGQC/
- Operating System(s): Platform Independent
- Programming language: R
- License: MIT
- RRID: SCR_022982.

## DATA AVAILABILITY

Test datasets can be found at https://github.com/LodovicoTerzi/HTGQC/tree/main/dataExample/. Both pre-formatted and unformatted test datasets can be found, along with a sample annotation dataset. Snapshots of the code are available in the GigaDB repository [13].

## DECLARATIONS

### Abbreviations

COO: Cell Of Origin; cpm: counts per million; FFPE: Formalin-Fixed Paraffin-Embedded; PCA: Principle Component Analysis; vst: variance stabilizing transformation.

### Ethics approval

No ethical approval was required.

## Competing Interests

The authors declare that they have no competing interests.

## Authors' contribution

LTDB conceived the study, wrote the manuscript, and developed HTGQC and shinyHTGQC. FG developed shinyHTGQC and wrote the manuscript. DR and FB contributed to the design of the work and manuscript revision. LC supervised the study and wrote the manuscript. All authors read and approved the final manuscript.

## Funding

No external funding was used.

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
