## [Reviewer Report]

Reviewer name and names of any other individual's who aided in reviewerAlessandro Lagana'Do you understand and agree to our policy of having open and named reviews, and having your review included with the published manuscript. (If no, please inform the editor that you cannot review this manuscript.)YesIs the language of sufficient quality?YesPlease add additional comments on language quality to clarify if neededIs there a clear statement of need explaining what problems the software is designed to solve and who the target audience is? YesAdditional CommentsIs the source code available, and has an appropriate Open Source Initiative license <a href="https://opensource.org/licenses" target="_blank">(https://opensource.org/licenses)</a> been assigned to the code?YesAdditional CommentsAs Open Source Software are there guidelines on how to contribute, report issues or seek support on the code?YesAdditional CommentsIs the code executable?YesAdditional CommentsThe shiny app fails with a generic error (An error has occurred. Check your logs or contact the app author for clarification) when generating the VST-based plots. The test was performed using the example data provided by the authors in the github page.Is installation/deployment sufficiently outlined in the paper and documentation, and does it proceed as outlined?YesAdditional CommentsIs the documentation provided clear and user friendly?YesAdditional CommentsIs there enough clear information in the documentation to install, run and test this tool, including information on where to seek help if required?YesAdditional CommentsThe shiny app is well documented. A small vignette with a tutorial in the github page for the R package would be helpful.Is there a clearly-stated list of dependencies, and is the core functionality of the software documented to a satisfactory level?NoAdditional CommentsThe authors do not provide a list of dependencies.Have any claims of performance been sufficiently tested and compared to other commonly-used packages? Not applicableAdditional CommentsThe tool appears to be the first for the task.Is test data available, either included with the submission or openly available via cited third party sources (e.g. accession numbers, data DOIs)?YesAdditional CommentsAre there (ideally real world) examples demonstrating use of the software? YesAdditional CommentsThere is example data in the github page. No real world application is described in the manuscript.Is automated testing used or are there manual steps described so that the functionality of the software can be verified?YesAdditional CommentsAny Additional Overall Comments to the AuthorThe package is rather simple but implements functionalities that are potentially useful for users of the HTG-EDGE-seq panels. While the functions implemented are sufficiently described, the authors should also clearly explain the gap filled by their tool, i.e. the current scenario for the analysis of this type of data: how do HTG-EDGE-seq users currently normalize and analyze their data? It would also be helpful to contextualize this new tool in an actual pipeline for data analysis: what other additional steps and tools would be required for a full analysis?RecommendationMinor Revisions

---

## [Reviewer Report]

Reviewer name and names of any other individual's who aided in reviewerChong TangDo you understand and agree to our policy of having open and named reviews, and having your review included with the published manuscript. (If no, please inform the editor that you cannot review this manuscript.)YesIs the language of sufficient quality?YesPlease add additional comments on language quality to clarify if neededThere are some typos. The author need to carefully address this typos.Is there a clear statement of need explaining what problems the software is designed to solve and who the target audience is? YesAdditional CommentsThe author developed HTGQC, a R package for the quality control of HTG Edge-Seq protocols, which may be important for medical researches.Is the source code available, and has an appropriate Open Source Initiative license <a href="https://opensource.org/licenses" target="_blank">(https://opensource.org/licenses)</a> been assigned to the code?YesAdditional CommentsAs Open Source Software are there guidelines on how to contribute, report issues or seek support on the code?YesAdditional CommentsIs the code executable?YesAdditional CommentsIs installation/deployment sufficiently outlined in the paper and documentation, and does it proceed as outlined?YesAdditional CommentsIs the documentation provided clear and user friendly?YesAdditional CommentsIs there enough clear information in the documentation to install, run and test this tool, including information on where to seek help if required?YesAdditional CommentsIs there a clearly-stated list of dependencies, and is the core functionality of the software documented to a satisfactory level?YesAdditional CommentsHave any claims of performance been sufficiently tested and compared to other commonly-used packages? NoAdditional CommentsActually I did not see the advantages and disadvantages of the software.Is test data available, either included with the submission or openly available via cited third party sources (e.g. accession numbers, data DOIs)?YesAdditional CommentsAre there (ideally real world) examples demonstrating use of the software? YesAdditional CommentsIs automated testing used or are there manual steps described so that the functionality of the software can be verified?NoAdditional CommentsAny Additional Overall Comments to the AuthorRecommendationMinor Revisions